# Green Synthesis of Silver Nanoparticles Using *Allium cepa* var. *Aggregatum* Natural Extract: Antibacterial and Cytotoxic Properties

**DOI:** 10.3390/nano12101725

**Published:** 2022-05-18

**Authors:** Jayashree Shanmugam, Manikandan Dhayalan, Mohammed Riyaz Savaas Umar, Mayakkannan Gopal, Moonis Ali Khan, Jesus Simal-Gandara, Antonio Cid-Samamed

**Affiliations:** 1Department of Biotechnology, Stella Maris College (Autonomous), Chennai 600086, Tamil Nadu, India; jayashreesanmugam2@gmail.com; 2Small Molecules and Drug Discovery Group, Anticancer Bioscience, Tianfu International Biotown Chengdu, Chengdu 610000, China; 3Department of Biotechnology, Islamiah College (Autonomous), Vaniyambadi 635752, Tamil Nadu, India; mriyaz4@gmail.com; 4Department of Marine Biotechnology, Amet University, Chennai 600040, Tamil Nadu, India; mayakkannang@ametuniv.ac.in; 5Chemistry Department, College of Science, King Saud University, Riyadh 11451, Saudi Arabia; mokhan@ksu.edu.sa; 6Nutrition and Bromatology Group, Department of Analytical Chemistry and Food Science, Faculty of Science, Universidade de Vigo, E-32004 Ourense, Spain; 7Physical Chemistry Department, Faculty of Sciences, University of Vigo, E-32004 Ourense, Spain

**Keywords:** silver nanoparticles, green synthesis, *Allium cepa* var. *Aggregatum* (shallot) extract, antibacterial activity, cytotoxicity

## Abstract

The chemical content of plant excerpts can be efficiently employed to reduce the metal ions to nanoparticles in the one-pot green production method. Here, green production of silver nanoparticles (AC-AgNPs) is performed by means of *Allium cepa* var. *Aggregatum* (shallot) extract as a stabilizer and reducer. The shape, size, and morphology of resultant AC-AgNPs are examined by optical spectroscopy analysis such as UV for nucleation and coalescence processes of the AC-AgNPs. Through FTIR functional group is determined and through DLS size is defined, it was confirmed that metallic AgNPs were successfully synthesized through the green synthesis route, and these results agreed well with the results obtained in the XRD pattern along with TEM spectroscopy, where the TEM images confirm the formation of sphere-like nanostructures along with SAED analysis. The chemical characterization is performed with XPS; the obtained molecular species in the materials are determined from the energy profile. Antioxidant activity of AC-AgNPs versus DPPH substrate is carried out. Antibacterial activity is well established against Gram-negative and Gram-positive organisms. Cell viability is accomplished, followed by an MTT assay, and a cytotoxicity assay of AC-AgNPs on MCF—7 cell lines is also carried out. Highlights: (1). This study highlights the eco-friendly synthesis of silver nanoparticles from *Allium cepa* var. *Aggregatum* Natural Extract. (2). The synthesized AC-AgNPs were characterized by UV-VIS, FT-IR, XRD, TEM, and XPS. (3). The synthesized nanoparticles were well dispersed in nature and the size range of 35 ± 8 nm. (4). The anti-candidal activity of biosynthesized silver nanoparticles was evaluated against the following Gram-Negative organisms: *Escherichia coli* (*E. coli*), and the following Gram-positive organisms: *Staphylococcus aureus* strains. The biosynthesized AC-AgNPs showed enhanced antiseptic features anti both Gram-positive and negative organisms. (5). Besides, the in vitro cytotoxic outcomes of AC-AgNPs were assessed versus MCF-7 cancerous cells, and the reduction in the feasibility of cancer cells was established via MTT assay, which suggests potential biomedical applications.

## 1. Introduction

Nanomaterials have offered practical solutions to many technological and environmental challenges, such as solar energy conversion, wastewater treatment [1], catalysis, electronics, optics, biosensors, drug delivery, and medical devices [1,2,3,4,5,6]. However, the toxicological assessment of these engineered products has questioned their wide range of usage. Thus, efforts are being made to develop sustainable, cost-effective, non-toxic, and productive synthesis routes for nanomaterials. The green methodology is a performance for the manageable manufacture of nanomaterials with a well-defined shape and size [7]. In recent studies, metal-based nanoparticles have been reported to offer promising potential in biomedical applications. Unique properties of the nanoparticles have been inculcated in the biosensor, nanocatalyst, antimicrobial, pharmaceutical, and environmental applications. 

The various advantages of nanomaterials are improving the bioavailability by means of enhancing the aqueous solubility and by targeting the drug to a specific location in the body to its site of action, and the disadvantages of nanotechnology are that it is costly, its development cost is high, and manufacturing of nanomaterials is also tricky. The green synthesis of silver nanoparticles with various plant extracts has shown potent antibiofilm, antibacterial, antioxidant, anticancer, and other biochemical activities. Hence, the green synthesis of nanoparticles is superior to physical and chemical ones. It is a simple, non-toxic, with no disposal issues, cost-effective production, and environmentally friendly synthesis method. [8].

In the biological synthesis of nanoparticles by encapsulation, they become functionalized to deliver the compounds with better delivery properties due to the adherence to different functional groups of the secondary metabolites from plants, which ensures better therapeutic efficacy. Hence, the biogenic synthesis of nanoparticles via biomolecules enhances their properties and promotes nanomaterial stabilization. [9,10,11].

However, the mechanism of the biogenic synthesis of nanoparticles is still not fully understood. Recent studies are being performed to improve the understanding of the biological processes behind nanoparticle synthesis with the unprecedented potential of new types of applications using the unearthed abilities of microorganisms. Microorganisms can synthesize many unique nanostructures, which has led researchers to become more interested in using these microorganisms to synthesize different nanostructures for various applications. There are a wide variety of microorganisms, such as algae, fungi, and bacteria, which are being used to create nanosized particles that react mainly with different metal precursors to produce nanoparticles. During the start of the synthesis process, the nucleation of HAuCl ions occurs, leading to the formation of nanoclusters using the electrostatic interface; then, gradually, they move across the cell wall of the microorganisms. The exact mechanism behind the nanoparticle synthesis extracellularly includes the involvement of reduction of ions via enzymes through the aggregation of metal ions such as Ag and Au on the cell surface.

Ameen et al. used *Cupriavidas* sp. for extracellular synthesis of silver NPs, as the nitrate reductase enzyme, where the silver ions are trapped on the bacterial cell surface, and the subsequent enzyme reduces them to silver NPs. Nowadays, other metal ions such as gold, titanium oxide, silver, zinc, copper, etc., are used in a wide range of synthesized NPs. The following Table 1 reports the results of previous research that highlight the importance of similar research. These nanoparticles were generated using a green synthesis methodology.

Shallot (*Allium ascalonicum*) is frequently used in customary medicine to treat fever, cold, and cough [12]. It has been reported that shallot extract is a rich font of flavonoids and polyphenols, for example, gallic acid, quercetin, kaempferol, eriodyctiol, apigenin, tannic acid, and isoquercetin [13,14]. Previous studies have reported the synergetic use of ultrasonication and chemical reduction to produce AgNPs and layered AgNPs working with chitosan and gallic acid (GC-AgNPs) [15]. Katherine et al. concluded that GC-AgNPs showed strong antibacterial assets and poisonous outcomes versus *E. coli*. Hence, a proposed ultrasound-assisted technique was fast, simple, and economical to produce GC-AgNPs as a promising antimicrobial agent.

Taking it into account, we have focused our experiment on synthesizing AgNPs by shallot extract and ultrasonication, where ultrasonication aids the chemical reduction led by components of the natural extract, for example, gallic acid, quercetin, kaempferol, eriodictyol, apigenin, tannic acid, and isoquercetin. 

As the aqueous leaf extract of plant extract was added to silver nitrate solution, the color of the solution changed from pale yellow to reddish-brown after being sonicated for 10 min at room temperature because of the process of reduction of Ag + to Ag° nanoparticles, and this indicated the biosynthesis of plant extract AC-AgNPs. The UV−vis spectra of the plant extract AC-AgNPs and the aqueous plant extract give a sharp peak from 410 to 470 nm.

The synthesized AC-AgNPs were characterized by UV-VIS, FT-IR, XRD, TEM, and XPS. In this investigation, the application of AC-AgNPs as an antimicrobial agent was verified against designated the following negative organisms: *Escherichia coli* (*E. coli*), and the following Gram-positive organisms: *Staphylococcus aureus*, on an agar plate and liquid medium. The results exhibited that the tested bacteria might be entirely inhibited by ac-AgNPs. The inhibition of bacteria growth was affected by the concentration of AC-AgNPs, and bacteria used in the trials. The green synthesized AC-AgNPs in this study can inhibit the high concentration of bacteria. This inhibition indicated that AgNPs showed a tremendous antimicrobial effect as the high concentration of bacteria used in this study [16]. The antioxidant activity and anticancer efficacy of the synthesized AC-AgNPs were evaluated to provide a green alternative for potential biomedical application.

**Table 1 nanomaterials-12-01725-t001:** Highlights the significance of the nanoparticles generated by green synthesis methods.

S.NO	Author/Citation	Route of Synthesis	Metal Nanoparticles	Size of the NPs	Antibacterial Activity	In Vitro Studies
1	Pradeep Kumar et al. (2020) [17]	Green Synthesis	MZ-AgNPs	84 nm	MIC 50 µg/mL ZOI: 16.66 ± 0.57 mm 14.6 ± 0.57 mm @*Staphylococcus aureus*, *Salmonella typhii*	Red blood cells (RBC) and mammalian cells, HEK293 cells: Cytocompatibility *RBC count = 73%. *HEK293= 37%
2	Jung-Kul Lee et al. (2019) [18]	Green Synthesis	Ag-GSiO_2_ NPs	20 nm	MIC 30 µg/mL @*Staphylococcus* *aureus* and *Escherichia coli*	-
3	Jung-Kul Lee et al. (2017) [19]	Green Synthesis	CELE-AgNPs	20–50 nm	All microorganism similiar to the following: *B. cereus*, *S. aureus*, *E. coli*, *S. typhimurium*, *E. faecalis*, *C. tropicalis*, *C. krusei*, *C. lusitaniae*, *C. guilliemondii*, *P. chrysogenum*	L929 cell line: Concentrations of 0 μg/mL (control), 18 μg/mL (~IC50), and 25 μg/mL (~2 × IC50), Apoptotic cells did not show any adverse effects
4	M. Dhayalan et al. (2018) [20]	Green Synthesis	AgNPs	5–35 nm	ZOI: *Proteus vulgaris* and *Micrococcus luteus*, 11–19 mm and 11–13 mm. respectively	HepG2 cell lines AgNPs showed ~20% decrease in the tumor cells at 10 μg/mL concentration
5.	Current study	Green Synthesis	AgNPs	50 nm	*Escherichia coli* and *Staphylococcus aureus*, ZOI:20–33 mm	MCF-7 cell line, ~20% diminution in the cancer cells for 20 g/mL

## 2. Experimental Methods

### 2.1. Materials, Glassware, and Chemicals

Fresh *Allium cepa* var. *Aggregatum* (shallot) belonging to the *Liliaceae* family, was collected from the local market in Chennai, India. It was washed with deionized water (D.I.) and dried in the dark. Then, the dried sample was minced in an electric mixer. 

General laboratory glassware such as conical flasks, test tubes, beakers, measurement cylinders, and pipettes was acquired from Borosil Glass Works Ltd., Mumbai, India, and Riviera Glass Pvt. Ltd., Mumbai, India. Petri dishes, bottles, and culture tubes were purchased from Riviera Glass Pvt. Ltd., India. Microcentrifuge tubes, culture hipflasks, and culture loops were procured from Tarsons Products Pvt. Ltd., Mumbai, India.

Chemicals and reagents such as silver nitrate, 2,2-diphenyl-1-picrylhydrazyl (DPPH), chloroauric acid, methanol, dimethyl sulfoxide (DMSO), sulfuric acid, disodium hydrogen phosphate, ascorbic acid, sodium dihydrogen phosphate, acridine orange, ammonium molybdate, 3-(4,5-dimethylthiazol-2-yl)-2,5-diphenyltetrazolium bromide (MTT), ethylenediaminetetraacetic acid sodium salt (EDTA), and sodium bicarbonate, were purchased from Sisco Research Laboratory Chemicals, Mumbai, India. The following cell lines of MCF-7 were provided by National Centre for Cell Science (NCCS), Pune, India. Culture media: penicillin, fetal bovine serum (FBS), gentamycin, streptomycin, and amphotericin were provided by Sigma-Aldrich Chemicals, USA. Fetal calf serum, phenol red, L-glutamine, glucose, trypsin, nutrient agar (NA), Rosewell Park Memorial Institute Medium (RPMI), casein acid hydrolysate, barium chloride, beef heart infusion, agar, and soluble starch were provided by Himedia Laboratories, Pune, India.

### 2.2. Preparation of Aqueous Extract of Shallot

Briefly, in a 250 mL Erlenmeyer flask containing 100 mL of D.I. water, 50 g of finely ground shallot was suspended under constant stirring in a shaker at 150 rpm for an hour. Thereafter, the blend was filtered through Whatman No. 1 filter paper, and the extract gathered was kept in the freezer at 8 °C for later use [21].

### 2.3. Synthesis of Silver Nanoparticles

To produce silver nanoparticles (AC-AgNPs), 1.0 mL of shallot extract was mixed with 9.0 mL of 1 mM AgNO_3_ solution. The mixture was sonicated for 10 min at room temperature to obtain a solution that changed from pale yellow to reddish-brown because of the process of reduction of Ag + to Ag° nanoparticles. The change in mixture color to reddish-brown affirmed the accomplishment of AC-AgNPs synthesis [22].

### 2.4. Characterization Techniques

UV–visible spectroscopy (SV 210 UV double beam, ELICO, India) with 1 nm resolution was performed to screen the bio-reduction of silver ions. The AC-AgNPs size distribution and ζ-potential values were recorded by Dynamic Laser Scattering (DLS) analysis using a 532 nm laser beam at 25 °C, using typical cuvettes or DTS1060C Clear Disposable zeta cells. The measurements were made in triplicates, and average values have been reported [23]. The samples for transmission electron microscopic (TEM) (JEM-3100F JEOL 3010 instrument with a UHR polepiece) analyses were centrifuged for 15 min at 10,000 rpm and re-dispersed three times in D.I. water to eliminate the pollution and any boundless biomolecules [24]. The obtained pellet was dehydrated overnight in a hot-air oven at 65 °C and analyzed. The crystallinity of AC-AgNPs was analyzed by an X-ray diffractometer (XRD-SMART lab Rigaku, Japan) run at a current of 100 mA with Cu Kα radiation and a voltage of 30 kV. Selected area electron diffraction (SAED) analysis (TECNAI 20 G2, FEI make) was also performed during the study. Fourier transform infrared (FT-IR) spectroscopic analysis (Perkin Elmer spectrum RXI FTIR) was performed to recognize biomolecules in shallot extract that lead to the bio-reduction of AC-AgNPs. This process was accomplished earlier and later the reaction with metal precursor. The chemical characterization was achieved with X-ray photoemission spectroscopy (XPS) (VGESCALAB MKII) via monochromatic Mg Ka X-radiation, operating at a pressure higher than 10-6 Pa.2, 2-diphenyl-1-picrylhydrazyl (DPPH) radical scavenging activity.

Antioxidant activity of AC-AgNPs, as opposed to DPPH substratum, was accomplished following the approach described by Guha et al. [25] with few modifications. In total, 2 mL of AC-AgNPs were dispersed in varying amounts of methanol (40, 30, 20, and 10 μg/mL). Thus, 1 mL of DPPH dispersion (0.2 mM/mL methanol) was placed, and the substances were energetically blended. The blend was maintained in the dark during 40 min at 20 °C and incubated. Following incubation, the absorbance was determined at 517 nm by means of a UV–visible spectrophotometer using methanol as blank. This routine used ascorbic acid as a standard. 

The free radical scavenging activity (*RSA*) of the extract was determined as follows:(1)$$RSA\;\left(\%\right)=\frac{OD_{Control}-OD_{Treated\;sample}}{OD_{Control}}\times 100\;$$
where *OD_Control_* is the absorbance (optical density) of the control (DPPH), *OD_Treated sample_* is the absorbance of the treated sample [26]. The tests were carried out in triplicate, and mean values were reported.

### 2.5. Antimicrobial Activity Assay

The agar well diffusion method was employed to assess antibacterial activity assay of AC-AgNPs [27]. The bacterial culture gram-negative *Escherichia coli* MTCC40 pEx2717 and gram-positive *Staphylococcus aureus* MTCC 87 Mkv2011 are used for antibacterial activity assay. The overnight cultures of both bacteria were spread and plated in the nutrient agar medium (pH 6.8), were employed for the activity, and incubated overnight at 37 °C. After overnight incubation, fully grown bacterial plates were chosen, and in those selected plates, 6 mm diameter wells were prepared by gel-hole punch, and 50 μL of AC-AgNPs were laden inside the wells. The plates were then incubated overnight at 37 °C for 24 h, and the zone of inhibition (ZOI) shaped nearby the wells was calculated. The test was made in triplicate, and the average values were documented. The outcomes of AC-AgNPs were correlated with the negative D.I. water and positive control with Ciprofloxacin 10 µg/mL [28]. The plates were incubated at 37 °C for 24 h, and the zone of inhibition (ZOI) shaped nearby the wells was calculated. The test was made in triplicate, and the average values were documented.

### 2.6. Minimum Inhibitory Concentration of Silver Nanoparticles 

The determination of MIC of the AC-AgNPs against *E. coli* and *S. auerus* was performed as per the recommendations of the Clinical Laboratory Standard Institute (CLSI 2008b). The bacteria were grown under aerobic conditions at 37 °C for 24 h in LB broth. The cell count was made with a hemocytometer, and a standardized cell suspension (1 × 105 cells/mL) was prepared. One hundred microliters of the cell suspension were dispensed in triplicate microtiter wells to which 100 μL suspension containing different concentrations of nanoparticles in LB medium was added and mixed well. A conventionally used antibacterial agent such as triclosan [29,30] was used to compare the antibacterial potential of AC-AgNPs. Triclosan was dissolved in dimethyl sulfoxide (DMSO) and diluted with sterile LB medium to obtain drug concentrations ranging from 0.0313 to 1024 μg/mL. The cells with either nanoparticles or conventionally used antibacterials were incubated at room temperature for 24 h, and the growth inhibition was measured spectrophotometrically at 600 nm (Powerserve XS Biotech, USA). Wells without AgNPs were used as the control. The MIC was calculated as the lowest amount of AgNPs that inhibited 50% growth of *E. coli* and *S. aureus* cells under the experimental conditions.

## 3. Results and Discussion

### 3.1. UV-Visible Spectrophotometer Analysis

The UV-vis spectra of a reaction mixture at room temperature (Figure 1a) and an ultrasonicated reaction mixture (Figure 1b) have been carried out to identify the differences between the synthesis methods employed.

The peak at 450 nm would correspond to the nucleation and coalescence processes of the AC-AgNPs. Considering the results obtained at room temperature (reaction time of 2 h), the reaction was carried out under ultrasonication. Kinetics for the formation of AgNPs in the presence and absence of sonication were observed in the light of UV–vis spectra. The absorption pattern suggested that the formation of AgNP started rapidly, and it became a significant amount within 10 min in the presence of sonication. While the formation of an appreciable number of AC-AgNPs requires at least 80 min without sonication [31]. 

AgNO_3_ + *Allium cepa var* Plant extract (flavonoids, terpenoids, alkaloids, polyphenols, alcohol, phenolic acids, antioxidants, vitamins).

(The AC-AgNPs nanoparticles are synthesized due to the electrostatic interaction between the functional groups of the respective constituent of plant extract and silver nitrate ions) [32]. Please, see Figure 1.

The peak attributable to the nucleation and aggregation processes was again observed at λ = 445 nm. Still, no experimental evidence of the pre-nucleation process appeared in this case, occurring in the reaction at room temperature.

In summary, shallot-mediated AC-AgNPs synthesis resulted in the appearance of a reddish-brown color under ambient conditions (after 2 h) and under ultrasonication in only 10 min, which is a reasonable time saving on the synthesis method. The SPR (surface plasmon resonance) band was identified at 444—437 nm. As the reduction reaction was very slow (prolonged) under room temperature, the ultrasonication increased the reaction rate (reaction time reduced to minutes instead of hours). The ultrasonicated sample was subjected to further characterization, frequency range (f) from 20 to 120 kHz.

### 3.2. Dynamic Light Scattering (DLS) Experiment and Transmission Electron Microscopy (TEM) Analysis

The constancy of the synthesized AC-AgNPs was analyzed using the DLS technique in the standings of dispensability and hydrodynamic measurements. DLS analyses were accomplished to clarify the mean average size (in diameter) and the polydispersity index (PDI) of the re-dispersed samples of AC-AgNPs. DLS data obtained indicated the average diameter of particles was 58.71 nm (PDI = 0.245) (Figure 2). The TEM images confirmed the spherical morphology of AC-AgNPs (Figure 3a). 

The polydispersity of the colloids is due to the multicomponent nature of AC extract that is liable for the reduction of Ag ions into their consistent metals. It is indispensable to look into the phytoconstituents and their nature for the complete consideration of the reduction of silver.

### 3.3. Morphological Analysis

HR-TEM images were obtained from the thorough investigation of AC-AgNPs samples. 

TEM micrographs and the SAED pattern of the biosynthesized AC-AgNPs were studied to determine the crystalline structure, morphology, and arrangement of the crystalline planes. The morphological features of the produced AC-AgNPs were characterized through TEM and illustrated in Figure 3a,b. However, the AC-AgNPs are dispersed; they seem overlapped in the images but not aggregated. This behavior is the case observed in the AC-AgNPs. 

The TEM images confirmed the formation of sphere-like nanostructures. The nanospheres synthesized through the green synthesis route were free from agglomeration. This behavior might be due to the synergic effect of the shallot extract and sonication process.

The TEM images agree with the results obtained in DLS. The stability of the produced AC-AgNPs was analyzed using the DLS technique in terms of dispersibility and hydrodynamic measurements.

The particle size distribution histogram of AC-AgNPs (Figure 3c) confirmed that the mean size of the produced silver NPs was 35 ± 8 nm. Additionally, the SAED pattern of AC-AgNPs was recorded (Figure 3d). The SAED pattern indicates well-defined diffraction rings. The diffraction ring indicated by Green corresponds to the (311) plane of metallic Ag. Therefore, from the SAED analysis, it was confirmed that metallic AC-AgNPs were successfully synthesized through the green synthesis route, and these results agreed well with the results obtained in the XRD pattern.

### 3.4. X-ray Diffraction Analysis

The crystal phase and arrangement of the produced AC-AgNPs have been investigated with XRD spectroscopy. The XRD analysis results, displayed in Figure 4, showed strong diffraction peaks. These peaks indicate that the prepared AC-AgNPs were crystalline. The diffraction peaks identified at an angle 2θ of 77.5°, 64.6°, 44.5°, and 38.3° agreed well with the (311), (220), (200), and (111) planes of cubic metallic Ag (JCPDS #: 04-0783) [33]. Moreover, no peaks related to AgO_2_, Ag_3_O_4,_ or other Ag molecules were observed in the diffraction pattern, revealing that the prepared material has a high degree of phase purity. The mean crystal size of the metallic Ag was estimated via the Debye−Scherrer equation [34,35,36] and was 38.6 nm.

### 3.5. X-ray Photoelectron Spectroscopy Analysis

The elements of Ag, O, C, and N were detected, as shown in Figure 5a–f. It can be observed that two peaks of Ag occurred at 371 eV and 377 eV, which correspond to Ag 3d5/2 and 3d3/2 binding energies, respectively. To verify the integration of nitrogen atoms inside the AC-AgNPs and to categorize the carbon-nitrogen species produced through the production route, XPS analysis was carried out. The XPS analysis of the scan spectrum of AC-AgNPs (Figure 5) revealed the presence of peaks ascribed to Ag, C, O, Ca, and N. It may indicate that silver atoms interact with N or O of the lead samples. The results obtained from the XPS analysis are illustrated in Figure 5.

Figure 5b shows the deconvoluted silver, which showed two peaks between 371 and 377 eV for Ag 3d5/2 and 3d3/2 coupling energies, respectively. The other peaks at 534, 287, 351, 354, and 403 eV correspond to O1s, C1s, Ca, and N1s, respectively. The O1s crests (in the 529.0–537.0 eV region), appearing from the oxygen-carbon species or water assimilated onto the surface of the AC-AgNPs, are shown in Figure 5c. These contributions, centered at 534.2 ± 0.2 eV, were associated with N-C-O and adsorbed H_2_O. This contribution centered at 287 eV corresponds to sp^2^ (C=C) and C-C [37]. The peaks between 351 and 354 correspond to Ca 2p^3/2^ [38], attributed to the carboxylate group.

### 3.6. Fourier Transform Infrared Spectroscopy Analysis

FTIR studies were accomplished for both the extract and reaction mixtures to determine the role of possible biomolecules in the stabilizing and reduction reactions of AC-AgNPs. Following the FTIR examination, results are shown in Figure 6.

The FTIR spectrum of *Allium cepa* var. *Aggregatum* natural extract and the biosynthesized AC-AgNPs have the same signature and showed several absorbance shoulders at 3449.50 cm^−1^, 2101.76 cm^−1^, 1638.07 cm^−1^, 1057.79 cm^−1^, and 641.05 cm^−1^ (Figure 6). The shoulders at 3449.50 cm^−1^ contributed to the N-H stretching of protein, the O-H widening of carbohydrates, and water. The shoulder at 2101.76 cm^−1^ was ascribed to the vibration of C≡C Terminal alkyne (monosubstituted). The bands obtained at 1638.07 cm^−1^ corresponded to the C–C stretching of phenyl present within the polyphenol components in the shallot extract; this structure was liable for the reduction of Ag^+^ ion to Ag° and stabilized the biosynthesized silver NPs. The peak at 1057.79 cm^−1^ was caused by the C-OH stretching of carbohydrates. The absorption band at 641.05 cm^−1^ was affixed to C-H out of the plane bend of aromatic phenols [39]. Besides, the –OH group on carbohydrate residues aided avoid aggregation or stabilizer on the surface of biosynthesized AC-AgNPs [40].

### 3.7. DPPH Asssay (2,2-diphenyl-1-picrylhydrazyl) Assay

The free radical scavenging activity in vitro was examined for various concentrations of AC-AgNPs from 10 to 40 μg/mL. They were compared to ascorbic acid [41] as a standard for antioxidant assay. DPPH is a constant free radical with a red color, which flashes yellow when scavenged. The DPPH assay utilizes this character to demonstrate free radical scavenging activity. The scavenging reaction between (DPPH) and an antioxidant (H-A) can be written as follows: (2)$$\left(\mathrm{DPPH}\right)+\left(H-A\right)\;\to \;\mathrm{DPPH}-H\;+\left(A\right)$$

The antioxidant responds to DPPH and reduces it to DPPH-H, and the absorbance decreases. The degree of staining points out the scavenging latent of the antioxidant compounds or extracts in terms of hydrogen donating ability.

The DPPH assay showed the enhanced radical scavenging activity of AC-AgNPs since these nanoparticles showed an excellent antioxidant activity [42]. Besides, AC-AgNPs showed higher antioxidant activity than ascorbic acid for all tested concentrations (Figure 7). 

### 3.8. Antibacterial Activity

The Agar well diffusion routine carried out the antibacterial activity assay of AC-AgNPs [43]. Antibacterial activity is well established comparing with Gram-negative and Gram-positive bacteria. As observed in Figure 8, both bacterial strains showed the mean value of ZOI with slight differences.

AC-AgNPs pose a greater volume surface-to-volume ratio related to their bulk equivalent. Thus, AC-AgNPs ease some exchanges with the bacteriological exteriors, so antibacterial activity increases [44]. Containing constituents of bacterial cells (sulfur and phosphorus), when interacting with AC-AgNPs, were reported to initiate cell death by assaulting the cell division and breathing chain [45]. Compared with other biosynthesized AC-AgNPs, the minimum value of ZOI matches with previous AC-AgNPs synthesized with Tulsi extract or quercetin (14 nm) [46], and the mean value of 20.33 nm was higher, so it can be deduced that silver NPs (with shallot extract and ultra-sonication method) show the same or higher antibacterial potential than previous research biosynthetic methods for AC-AgNPs [47]. 

### 3.9. Minimum Inhibitory Concentration of Silver Nanoparticles

A minimum inhibitory concentration (MIC) test was performed to check the minimum concentration of biogenic AgNPs against bacterial strains. The MIC test reveals the existence of antibacterial activity at different concentration levels. In the present investigation, the MIC test value of biogenic silver nanoparticles against both the strains of Escherichia coli MTCC40 pEx2717 and Staphylococcus aureus MTCC 87 Mkv2011 was calculated as 0.27 and 0.97 µg/mL, respectively, and triclosan was calculated as between 0.5 and 2 µg/mL, respectively (Figure 9). Interestingly, the IC50 value was very much lower than the IC50 value of the commercial drug. AgNPs synthesized using different plant extracts were effective against various bacteria. For example, AgNPs biosynthesized by the Calotropis gigantea leaf extract are against C. albicans with a MIC of 50 μg/mL [48]. In contrast, biosynthesized AgNPs using Artemisia annua showed MIC against C. albicans, C. tropicalis, and C. glabrata that ranged between 80 and 120 μg/mL. Thus, AgNPs produced by different approaches and species were reported to show antibacterial activity at different MIC levels depending on their size, shape, and surface modification [49].

The size of the nanoparticles plays an essential role in antimicrobial activity. It has been reported that the size and shape of metallic nanoparticles influence their chemical, optical, and thermal properties.

### 3.10. (3-(4,5-dimethylthiazol-2-yl)-2,5-diphenyltetrazolium bromide) Tetrazolium (MTT) Assay

The cytotoxic studies of AC-AgNPs were investigated in MCF cells by using an MTT assay. The viability of MCF-7 cell lines was found to decrease with an increase in the concentration of AC-AgNPs in a dose-dependent manner (1 µg/mL–100 µg/mL). The graph shows the inhibition of cell proliferation in MCF-7 cells in a dose-dependent manner (Figure 10 and Figure 11). The MCF-7 cell proliferation was considerably lower when compared to untreated cells. After the treatments for 24 h and 48 h higher concentrations (50 µg/mL and 100 µg/mL) killed more than 70% of cancer cells. The maximum inhibitions of cell proliferations were obtained at the concentration of (20 µg/mL) after 24 h and 48 h (Figure 10). The AC-AgNPs concentration at 20 µg/mL exhibited 75% cell viability and the 50 µg/mL concentration showed 60% cell viability. It was observed from the results that AC-AgNPs have imparted a remarkable effect on MCF-7 cells by creating cell damage, which agrees well with the literature [50]. NiO nanoparticles synthesized using *Aegle marmelos* leaf extract were shown to have a cytotoxicity effect on A459 cell lines. The cell viability was observed to be 20% when treated with 100 µg/mL of NiO nanoparticles and increased gradually to a maximum of 64.6% at a concentration of 7.8 µg/mL [51]. The exact mechanism for the cytotoxic effect of NP is not clearly known, but a few reports suggest that the metal ion that is released from the nanometal oxides plays a key role in cytotoxicity [52,53].

Mnayer et al. [7] carried out a GC-MS analysis of the shallot essential oil (EO), and they reported that 42 compounds that signify more than 70.29% of the total EO were recognized. The major components were methyl propyl disulfide (3.26%), methyl 1-propenyl propyl disulfide (4.57%), propyl trisulfide (9.20%), dipropyl trisulfide (11.14%), and dipropyl disulfide (15.17%).

## 4. Conclusions

A one-step ultrasonication-assisted (for 10 min) green synthesis process of AC-AgNPs using *Allium cepa* var. *Aggregatum* natural essence as a stabilizer and reducer under room temperature conditions was proposed. The produced AC-AgNPs displayed a strong absorption maximum at 450 nm, dependent on the resultant particles’ morphology, shape, and size. The mean particle size of manufactured silver NPs ranged between 27 and 43 nm in diameter. XRD analysis confirmed the metallic structure of AC-AgNPs and agreed well with the (311), (220), (200), and (111) planes of cubic metallic Ag. XPS analysis of scan spectra of silver NPs proves the strong shoulders ascribed to N, Ca, O, C, and Ag, implying the occurrence of AC-AgNPs coated by the coating agent of shallot extract. Surrounding media can meaningfully change the shape and size of manufactured AC-AgNPs.

Consequently, the required shape and size of silver NPs aimed at several purposes can be performed simply by means of this green method. Moreover, the DPPH assay displayed improved radical scavenging activity of AC-AgNPs since these AC-AgNPs showed excellent antioxidant activity. The biosynthesized AC-AgNPs showed enhanced antimicrobial features against both Gram-positive and negative organisms. Besides, the in vitro cytotoxic outcomes of silver NPs were assessed versus MCF–7 cancerous cells, and the reduction in the feasibility of cancer cells was established via MTT assay, which suggests potential biomedical applications.

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
