# Peer review of "Green Synthesis of Silver Nanoparticles Using Allium cepa var. Aggregatum Natural Extract: Antibacterial and Cytotoxic Properties"

_nanomaterials, 2022, doi:10.3390/nano12101725_

Round 1

Reviewer 1 Report

The manuscript by Jayashree et al. “Green Synthesis of Silver Nanoparticles Using Allium Cepa var. Aggregatum Natural Extract: Antibacterial and Cytotoxic Properties” requires revision to address major concerns before its publication. 

Comments

  1. The authors should cross-verify all abbreviations used in the abstract as well as in the main text, separately. Abbreviations should be consistent throughout the manuscript i.e. “nanoparticles” as “NPs”.
  1. The introduction should be more briefly elaborated with a piece of basic information and citations such as i) the various advantages and disadvantages of nanomaterials; ii) the importance of biological synthesis of nanoparticles over physio-chemical synthesis; iii) various approaches of bio-based nanoparticles synthesis and challenges, iv) bio-mechanisms of NPs synthesis, and v) various anti-microbial agents and suitability of biosynthesized nanomaterials as an anti-microbial application i.e. doi: 10.1007/s12088-020-00889-0; doi: 10.4014/jmb.1610.10019; doi: 10.1016/j.chemosphere.2022.134497;doi: 10.1016/j.biortech.2021.124737; doi: 10.1002/bab.2235; and doi: 10.1007/s12088-019-00812-2.
  1. Please highlight the novelty and significance of the present study.
  2. Results and discussion, the discussion section is a week, please improve it (each section) significantly with citations i.e. significance or justification of finding.
  3. Please provide the comparison Tables based on a similar study to highlight the significance of this article.
  4. Illustrate the mechanism for the synthesis of NPs and how it acts as antimicrobial?
  5. The figures quality should be improved and please combine a few figures.

Author Response

Reviewer: 1

Author response to reviewer

The manuscript by Jayashree et al. “Green Synthesis of Silver Nanoparticles Using Allium Cepa var. Aggregatum Natural Extract: Antibacterial and Cytotoxic Properties” requires revision to address major concerns before its publication. 

 Comments

  1. The authors should cross-verify all abbreviations used in the abstract as well as in the main text, separately. Abbreviations should be consistent throughout the manuscript i.e. “nanoparticles” as “NPs”. 

Ans: (As per reviewer comments, we had carefully verified and made all abbreviations used in the abstract and in the main text.)

  1. The introduction should be more briefly elaborated with a piece of basic information and citations such as
  1. i) the various advantages and disadvantages of nanomaterials;
  2. ii) the importance of biological synthesis of nanoparticles over physio-chemical

   synthesis.

iii) various approaches of bio-based nanoparticles synthesis and challenges

  1. iv) bio-mechanisms of NPs synthesis, and
  2. v) Various anti-microbial agents and suitability of biosynthesized nanomaterials as an

     anti-microbial application

i.e. doi: 10.1007/s12088-020-00889-0; doi: 10.4014/jmb.1610.10019; doi: 10.1016/j.chemosphere.2022.134497;doi: 10.1016/j.biortech.2021.124737; doi: 10.1002/bab.2235; and doi: 10.1007/s12088-019-00812-2.

Ans:  (i) The various advantages of nanomaterials are like improving the bioavailability by means of enhancing the aqueous solubility and also by targeting the drug to a specific location in the body to its site of action, and the disadvantages of nanotechnology as it is costly and its development cost is high, and manufacturing of nanomaterial is also tricky.

(ii) The green synthesis of silver nanoparticles with various plant extracts has shown a potent antibiofilm, antibacterial, antioxidant, anticancer, and other biochemical activities. Hence, the green synthesis of nanoparticles is superior to physical and chemical ones. It is a simple, non-toxic, with no disposal issues, cost-effective production, and environment-friendly synthesis method. [8-10]

(iii) In a biological synthesis of nanoparticles by encapsulation, they get functionalized to deliver the compounds with better deliver properties due to the adherence to different functional groups of the secondary metabolites from plants which ensures better therapeutic efficacy. Hence, biogenic synthesis of nanoparticles via biomolecules enhances its properties and promotes nanomaterials stabilization. [7, 9-10].

(iv) The biogenic synthesis of nanoparticle mechanisms is still not fully understood. Recent studies are being done to improve the understanding of the biological processes behind nanoparticle synthesis with the unprecedented potential of new types of applications using the unearthed abilities of microorganisms. Microorganisms can synthesize many unique nanostructures, which has led researchers to get more interested in using these microorganisms to synthesize different nanostructures for various applications. There are a wide variety of microorganisms, such as algae, fungi, and bacteria, which are being used to create a nanosized particle that reacts mainly with different metal precursors to produce nanoparticles. During the start of the synthesis process, the nucleation of HAuCl ions occurs, leading to the formation of nanoclusters using the electrostatic interface; then, gradually, they move across the cell wall of the microorganisms. The exact mechanism behind the nanoparticle synthesis extracellularly includes the involvement of reduction of ions via enzymes through the aggregation of metal ions like Ag and Au on the cell surface.

(v) Ameen et al. used Cupriavidas sp. for extracellular synthesis of silver NPs [29], as the nitrate reductase enzyme where the silver ions are trapped on the bacterial cell surface, and the subsequent enzyme reduces them to silver NPs. Nowadays, other metal ions like gold, titanium oxide, silver, zinc, copper, etc., are used in a wide range of synthesized NPs. The following table 1 highlights the significance of the similar research previously reported.

  1. Please highlight the novelty and significance of the present study.

Ans:

  1. This study highlights the eco-friendly synthesis of silver nanoparticles from Allium cepa var. aggregatum Natural Extract.
  2. The synthesized AC-AgNPs were characterized by UV-VIS, FT-IR, XRD, TEM, and XPS.
  3. The synthesized nanoparticles were well dispersed in nature and the size range of 35±8nm.
  4. Anti-candidal activity of biosynthesized silver nanoparticles was evaluated against Gram-Negative organisms: Escherichia coli (E.coli), and Gram-positive organisms: Staphylococcus aureus strains. The biosynthesized AC-AgNPs showed enhanced antiseptic features anti both Gram-positive and negative organisms.
  5. Besides, the in vitro cytotoxic outcomes of AC-AgNPs were assessed versus MCF-7 cancerous cells, and the reduction in the feasibility of cancer cells was established via MTT assay, which suggests potential biomedical applications.

  1. Results and discussion, the discussion section is a weak, please improve it (each section)

Ans: We have thoroughly researched the reviewer's ideas and questions and have improved the results and discussion, discussion areas and tagged the improved areas.

  1. Significantly with citations i.e. significance or justification of finding.

Ans: The significance and justification of the finding have been revised with recent citations.

  1. Please provide the comparison Tables based on a similar study to highlight the significance of this article.

ANS:

S.NO

Author/Citation

Route of synthesis

Metal Nanoparticles

Size of the NPs

Antibacterial activity

In Vitro studies 

1

Pradeep Kumar et al (2020)

Green Synthesis

MZ-AgNPs

84 nm

MIC 50µg/ml

ZOI: 16.66 ± 0.57 mm

14.6 ± 0.57 mm

@Staphylococcus aureus, Salmonella typhii

Red blood cells (RBC) and mammalian cells, HEK293 cells: Cytocompatibility *RBC count = 73%.

*HEK293= 37%

2

Jung-Kul Lee et al (2019)

Green Synthesis

Ag-GSiO2 NPs

20 nm

MIC 30 µg/ml

@Staphylococcus

aureus and Escherichia coli

--

3

Jung-Kul Lee et al (2017)

Green Synthesis

CELE-AgNPs

20 -50 nm

All microorganism like:B.cereus,S.aureus,E.coli,S.typhimurium,E.faecalis,C.tropicalis,C.krusei,C.lusitaniae,C.guilliemondii,P.chrysogenum

L929 cell line: Concentrations of 0 μg/ml (control), 18 μg/ml (~IC50), and 25 μg/ml (~2 × IC50), Apoptotic cells did not show any adverse effects

4

M. Dhayalan et al. (2018)

Green Synthesis

AgNPs

5-35 nm

ZOI: Proteus vulgaris and Micrococcus luteus, 11-19 mm and 11–13 mm, respectively

HepG2 cell lines AgNPs showed ~20% decrease in the tumor cells at 10 μg/mL concentration

5.

Current study

Green Synthesis

AgNPs

50 nm

Escherichia coli and Staphylococcus aureus , ZOI:20- 33 mm

MCF-7 cell line , ~20% diminution in the cancer cells for 20 μg/mL

  1. Illustrate the mechanism for the synthesis of NPs and how it acts as antimicrobial?

Ans: As the aqueous leaf extract of plant extract was added to silver nitrate solution, the color of the solution changed from pale yellow to reddish-brown after sonicated for 10 min at room temperature because of the process of reduction of Ag+ to Ag° nanoparticles, and this indicated the biosynthesis of plant extract AC-AgNPs. UV−vis spectra of the plant extract AC-AgNPs and the aqueous plant extract give a sharp peak at near 410 to 470 nm.

In this investigation, the application of AC-AgNPs as an antimicrobial agent was verified against designated Negative organisms: Escherichia coli (E. coli) Gram-positive organisms: Staphylococcus aureus, on an agar plate and liquid medium. The results exhibited that the tested bacteria might be entirely inhibited by ac-AgNPs. The inhibition of bacteria growth was affected by the concentration of AC-AgNPs and bacteria used in the trials. The green synthesized AC-AgNPs in this study can inhibit the high concentration of bacteria. This inhibition indicated that AgNPs showed a tremendous antimicrobial effect as the high concentration of bacteria used in this study [15].

  1. The figures quality should be improved and please combine a few figures.

Ans: We incorporated the images with improved quality in the revised manuscript.

Reviewer 2 Report

In this paper  a one-step ultrasonication-assisted synthesis of AgNPs  using plant extract Allium cepa var. aggregatum as a stabilizer and reducing under room temperature conditions was discribed. The produced AgNPs has 450 nm peak. The average particle diameter of AgNPs was about 27 and 43 nm. XRD analysis confirmed the metallic structure of AgNPs and XPS spectra of silver NPs reveal the strong shoulders ascribed to N, Ca, O, C, and Ag attributed to AgNPs coated by the coating agent of shallot extract. Antioxidant activity of the DPPH assay illustrated improved radical scavenging activity of AgNPs. The synthesized AgNPs showed some antiseptic activity for Gram-positive and negative organisms. the in vitro cytotoxic activity of AgNPs were investigated aganist MCF–7 cancerous cells.

There are several comments to improve the quality.

 A small peak in UV spectra was observed around 350 nm. I think it is artefact of the instrument.

Page 2. Lines 42-43. “The green methodology is a performance for the manageable manufacture of nanomaterials with well-defined shape and size”. It seems to me that it would be better to indicate the appropriate ref.

Page 2. Lines 59-60. “Fresh vegetable Allium cepa var. aggregatum (shallot) was collected from the local market in Chennai, India.”. Please indicate the plant variety

Page 3. Lines 73-79. The bacterial strains used to assess the antimicrobial activity of AgNPs are not indicated.

Page 3. Lines 88-89. “The change in mixture color to reddish-brown affirmed the accomplishment of AgNPs synthesis”. It should be confirmed by the appropriate ref.

Lines 148-150. shallot-mediated AgNPs synthesis resulted in the appearance of reddish- brown color under ambient conditions (after 2 h) and under ultrasonication, in only 10 minutes, which is a reasonable time saving on synthesis method. Please prove that all silver ions were converted to AgNPs

Lines 212-214 These contributions, centered at 534.2 ± 0.2 eV, were associated with N-C-O and adsorbed H2O. This contribution centered at 287 eV correspond to sp2 (C=C) and C-C20. The peaks at 351 and 354 correspond to Ca 2p3/221. It is just reporting what kind of compounds it is attributed to?

The shoulder at 2101.76 cm-1 was ascribed to the vibration of C=N. It is not correct! Please correct.

Lines 233-237 The free radical scavenging activity in vitro was examined for various concentrations of AgNPs from 10 to 40 mg/mL. They were compared to ascorbic acid24 as a standard for antioxidant assay. DPPH assay showed….. AgNPs since these nanoparticles showed an excellent antioxidant activity 25. AgNPs showed higher antioxidant activity than ascorbic acid for all tested concentrations. No control experiment was done it is not appropriate standard antioxidant compound.  Please explain mechanism of antioxidant activity?

Page 11. Line 248. “Figure 8. Antibacterial activity”. The title of the figure should indicate against which bacterial strains the antimicrobial efficacy was determined. This information is also missing from the text.

Page 11. Lines 253-254. “Compared with other biosynthesized AgNPs, the mini-253 mum value of ZOI matches with previous AgNPs synthesized with tulsi extract or quer-254 cetin (14 nm)”. Please indicate the appropriate ref.

Figure 8. Antibacterial activity What does it mean A  and B What was the control sample?

Lines 253-254 Compared with other biosynthesized AgNPs, the minimum value of ZOI matches with previous AgNPs synthesized with tulsi extract or quercetin (14 nm), and the mean value of 20.33 nm was higher, so it can be deduced that silver NPs (with shallot extract and ultra-sonication method) shows the same or higher antibacterial potential that previous research biosynthetic methods for AgNPs Reference is missed.

Lines 253-254 The obtained results determine that AgNPs displayed ~20% diminution in the cancer cells for 20 mg/mL (Fig. 9). Hence, these biosynthesized AgNPs using shallot natural extract were good anticancer instruments by reducing the gradual cancer cells growth.  Inappropriate conclusion

Page 12. Figure 10 is not mentioned in the text.

Lines 271-272  Figure 10. MTT assay graph showing the percentage of viable cells using different concentrations  of AgNPs. Please add standard deviation

Author Response

Reviewer: 2

Author response to reviewer 2

In this paper a one-step ultrasonication-assisted synthesis of AgNPs using plant extract Allium cepa var. aggregatum as a stabilizer and reducing under room temperature conditions was described. The produced AgNPs has 450 nm peak. The average particle diameter of AgNPs was about 27 and 43 nm. XRD analysis confirmed the metallic structure of AgNPs and XPS spectra of silver NPs reveal the strong shoulders ascribed to N, Ca, O, C, and Ag attributed to AgNPs coated by the coating agent of shallot extract. Antioxidant activity of the DPPH assay illustrated improved radical scavenging activity of AgNPs. The synthesized AgNPs showed some antiseptic activity for Gram-positive and negative organisms. the in vitro cytotoxic activity of AgNPs were investigated aganist MCF–7 cancerous cells.

There are several comments to improve the quality.

 A small peak in UV spectra was observed around 350 nm. I think it is artefact of the instrument.

In response to Reviewer 2's request, we have removed the sentence: "A small peak was observed around 350 nm. The band described above would be ascribed to the previous nucleation processes during the reaction, while " after Figure 1.

  1. Page 2. Lines 42-43. “The green methodology is a performance for the manageable manufacture of nanomaterials with well-defined shape and size”. It seems to me that it would be better to indicate the appropriate ref

Ans: we added appropriate reference in the revised manuscript. The reference was included; it appears in the text, page 2, Ref NO:6: .Naumih M N, Peter MN, Green synthesis of nanomaterials from sustainable materials for biosensors and drug delivery, Sensors International,3, 100166(2022).)

  1. Page 2. Lines 59-60. “Fresh vegetable Allium cepa var. aggregatum (shallot) was collected from the local market in Chennai, India.”. Please indicate the plant variety

Ans:   (belongs to Liliaceae family) it was mentioned in the text page No : 4 , section 2. Experimental,2.1. Materials, glassware, and chemicals: first line)

  1. Page 3. Lines 73-79. The bacterial strains used to assess the antimicrobial activity of AgNPs are not indicated.

Ans:

Gram-Negative organisms: Escherichia coli (E coli)

Gram-positive organisms: Staphylococcus aureus,

Bacterial Culture: Bacterial cultures of Escherichia coli and Staphylococcus aureus were procured from Microbial Type Culture Collection (MTCC), Institute of Microbial Technology, Chandigarh.

It was incorporated into the revised manuscript.

  1. Page 3. Lines 88-89. “The change in mixture color to reddish-brown affirmed the accomplishment of AgNPs synthesis”. It should be confirmed by the appropriate ref.

Ans: The appropriate reference was included in the text: page No : 3 Ref: 15: Linlin Wang,Chen H,Longquan S, The antimicrobial activity of nanoparticles: present situation and prospects for the future.Int J Nanomedicine,12,1227–1249(2017).

  1. Lines 148-150. shallot-mediated AgNPs synthesis resulted in the appearance of reddish- brown color under ambient conditions (after 2 h) and under ultrasonication, in only 10 minutes, which is a reasonable time saving on synthesis method. Please prove that all silver ions were converted to AgNPs

Ans: we added appropriate reference in the revised manuscript, mentioned in the in the text  page No : 7 ,section 3.1: Ref:no;28 and 29:

  1. Swadhin K S, Pranesh C, Prasanta S, Santi P, Sinha B, Ultrasound assisted green synthesis of poly(vinyl alcohol) capped silver nanoparticles for the study of its antifilarial efficacy, Applied Surface Science, 288, (1 )625-632(2014).
  2. Manjamadha V P , Muthukumar K, Ultrasound assisted green synthesis of silver nanoparticles using weed plant, Bioprocess and Biosystems Engineering, 39, 401–411 (2016).))

Kinetics for the formation of AgNPs in the presence and absence of sonication was observed in the light of UV–vis spectra. The absorption pattern suggested that the formation of AgNP started rapidly, and it became a significant amount within 10 minutes in the presence of sonication. While the formation of an appreciable amount AC-AgNP required at least 80 minutes without sonication [28]

AgNO3 + Allium cepa var Plant extract (Flavonoids, terpenoids, alkaloids, polyphenols, alcohol, phenolic acids, antioxidants, vitamins)

(Electrostatic interaction between the functional groups of the respective constituent of plant extract and Ag+ ion) [29]. Please, see Scheme 1.

It was incorporated in the revised manuscript as text and as the following scheme:

  1. Lines 212-214 These contributions, centered at 534.2 ± 0.2 eV, were associated with N-C-O and adsorbed H2O. This contribution centered at 287 eV corresponds to sp2 (C=C) and C-C20. The peaks at 351 and 354 correspond to Ca 2p3/221. It is just reporting what kind of compounds it is attributed to?

Ans: observed at 534.2 ± 0.2 eV was assigned to the carboxylate group. It was incorporated in the revised manuscript as follows:

These contributions, centered at 534.2 ± 0.2 eV, were associated with N-C-O and adsorbed H2O. This contribution centered at 287 eV corresponds to sp2 (C=C) and C-C [34]. The peaks at 351 and 354 correspond to Ca 2p3/2 [35], attributed to the carboxylate group.

  1. The shoulder at 2101.76 cm-1 was ascribed to the vibration of C=N. It is not correct! Please correct.

Ans: 2101.76 cm-1 is C≡C Terminal alkyne (monosubstituted). It was incorporated in the revised manuscript as follows:

The shoulder at 2101.76 cm-1 was ascribed to the vibration of C ≡ C Terminal alkyne (monosubstituted).

  1. Lines 233-237. The free radical scavenging activity in vitro was examined for various concentrations of AgNPs from 10 to 40 mg/mL. They were compared to ascorbic acid24 as a standard for antioxidant assay. DPPH assay showed….. AgNPs since these nanoparticles showed an excellent antioxidant activity 25. AgNPs showed higher antioxidant activity than ascorbic acid for all tested concentrations. No control experiment was done it is not appropriate standard antioxidant compound.  Please explain mechanism of antioxidant activity?

Ans: It was incorporated in the revised manuscript as follows:

DPPH is a constant free radical with red color, which flashes yellow when scavenged. The DPPH assay usages this character to demonstrate free radical scavenging activity. The scavenging reaction between (DPPH) and an antioxidant (H-A) can be written as follows,

Antioxidant responds with DPPH and reduces it to DPPH-H, and the absorbance decreases. The degree of staining point out the scavenging latent of the antioxidant compounds or extracts in terms of hydrogen donating ability.

  1. Page 11. Line 248. “Figure 8. Antibacterial activity”. The title of the figure should indicate against which bacterial strains the antimicrobial efficacy was determined. This information is also missing from the text.

Ans:    Gram-Negative organisms: Escherichia coli (E coli)

Gram-positive organisms: Staphylococcus aureus,

Bacterial Culture: Bacterial cultures of Escherichia coli and Staphylococcus aureus were procured from Microbial Type Culture Collection (MTCC), Institute of Microbial Technology, Chandigarh.

It was incorporated into the revised manuscript.

  1. Page 11. Lines 253-254. “Compared with other biosynthesized AgNPs, the mini-253 mum value of ZOI matches with previous AgNPs synthesized with tulsi extract or quer-254 cetin (14 nm)”. Please indicate the appropriate ref.

Ans: we added appropriate reference in the revised manuscript. It was mentioned in the text  page No : 13 ,section 3.8: Paragraph 2  Ref:no;43. Siddhant J, Mohan SM, Medicinal Plant Leaf Extract and Pure Flavonoid Mediated Green Synthesis of Silver Nanoparticles and their Enhanced Antibacterial Property, Scientific Reports,7, 15867 (2017).

  1. Figure 8. Antibacterial activity What does it mean A  and B What was the control sample?

Ans:     Positive Control: Ciprofloxacin

Negative control: Distilled Water

Escherichia coli (E coli)

B Staphylococcus aureus, it was incorporated in the revised manuscript.

  1. Lines 253-254 Compared with other biosynthesized AgNPs, the minimum value of ZOI matches with previous AgNPs synthesized with tulsi extract or quercetin (14 nm), and the mean value of 20.33 nm was higher, so it can be deduced that silver NPs (with shallot extract and ultra-sonication method) shows the same or higher antibacterial potential that previous research biosynthetic methods for AgNPsReference is missed.

Ans: we added appropriate reference in the revised manuscript. It was mentioned in the text page No: 13, section 3.8: Paragraph 2  Ref:no;44. Sreekanth T V M, Nagajyothi P C, Muthuraman P, Enkhtaivan G, Vattikuti S V P, Tettey, C O, Doo H K, Jaesool S, Kisoo Y, Ultra-sonication-assisted silver nanoparticles using Panax ginseng root extract and their anti-cancer and antiviral activities, Journal of Photochemistry & Photobiology, B: Biology, 188,6-11(2018).

  1. Lines 253-254 The obtained results determine that AgNPs displayed ~20% diminution in the cancer cells for 20 mg/mL (Fig. 9). Hence, these biosynthesized AgNPs using shallot natural extract were good anticancer instruments by reducing the gradual cancer cells growth.Inappropriate conclusion.

Ans: The sentence was deleted in the revised manuscript.

  1. Page 12. Figure 10 is not mentioned in the text.

Ans: it is mentioned in the revised manuscript. It was mentioned in the text on the page No: 15, section 3.10: Paragraph 1).

  1. Lines 271-272  Figure 10. MTT assay graph showing the percentage of viable cells using different concentrations  of AgNPs. Please add standard deviation

 Ans: Standard deviation was incorporated in the revised manuscript.

Reviewer 3 Report

The manuscript nanomaterials-1692043 "Green synthesis of silver nanoparticles using Allium cepa var. aggregatum natural extract: antibacterial and cytotoxic properties" by Jayashreeet al. describes shallot (Allium cepa var. aggregatum) mediated green synthesis of silver nanoparticles (AgNPs) and the study of their biological activity. The synthesis was confirmed by XRD, UV, FTIR spectroscopy, TEM, DLS.

Questions and comments:

1) There are many studies on the green synthesis of silver nanoparticles and the study of their antibacterial activity. Authors should pay special attention to explaining the difference between their results and previous work of colleagues. Such a comparison, as well as a more detailed discussion and analysis of the results obtained, should be added.

2) The authors should strengthen the Introduction part about silver nanoparticles. The authors use references to studies 5-10 years ago. New articles on the design of silver nanoparticles, as well as their applications, should be added. For example, Int. J. Mol. Sci. 2020, 21(4), 1425; Nanomaterials 2022, 12(1), 31; Molecules 2021, 26(9), 2462.

3) What ultrasound frequency was used in AgNPs synthesis?

4) Figure 1a. Correspondence each line vs. the synthesis time is not clear. I recommend signing the lines. The curvature of the lines at 380-390 nm in the spectrum should be corrected.

5) Figure 8. A and B part were not signed. Information about which strains of bacteria were tested for should be indicated in the text of part 3.8.

6) Regarding biological part. Why were the MIC and MBC of the obtained silver nanoparticles not measured? What is the mechanism for improving the biological properties of AgNPs?

7) I believe that a comparison of the biological properties of silver nanoparticles without an organic component should be added (blank experiment).

8) What about the toxicity of the obtained nanoparticles on normal cells?

9) Minor comments.

- line 166. "Fig. 4 (a and b) " should be changed by "Fig. 3 (a and b) ".

- I recommend rechecking the manuscript for errors.

Author Response

Reviewer: 3

Author response to Reviewer 3

The manuscript nanomaterials-1692043 "Green synthesis of silver nanoparticles using Allium cepa var. aggregatum natural extract: antibacterial and cytotoxic properties" by Jayashreeet al. describes shallot (Allium cepa var. aggregatum) mediated green synthesis of silver nanoparticles (AgNPs) and the study of their biological activity. The synthesis was confirmed by XRD, UV, FTIR spectroscopy, TEM, DLS.

Questions and comments:

1) There are many studies on the green synthesis of silver nanoparticles and the study of their antibacterial activity. Authors should pay special attention to explaining the difference between their results and previous work of colleagues. Such a comparison, as well as a more detailed discussion and analysis of the results obtained, should be added.

Ans: Table comparison was added in the introduction part of the revised manuscript.

2) The authors should strengthen the Introduction part about silver nanoparticles. The authors use references to studies 5-10 years ago. New articles on the design of silver nanoparticles, as well as their applications, should be added. For example, Int. J. Mol. Sci. 202021(4), 1425; Nanomaterials 202212(1), 31; Molecules 202126(9), 2462.

Ans:  as requested by reviewer 3, the introduction section has been extended by adding the appropriate references in the text on page No: 3.

3) What ultrasound frequency was used in AgNPs synthesis?

Ans: an appropriate correction was made to frequency range (f) of 20 to 120 kHz, mentioned on page no:7, line no.271.

4) Figure 1a. Correspondence each line vs. the synthesis time is not clear. I recommend signing the lines. The curvature of the lines at 380-390 nm in the spectrum should be corrected.

Ans: appropriate corrections are made.

5) Figure 8. A and B part were not signed. Information about which strains of bacteria were tested for should be indicated in the text of part 3.8.

Ans: Gram-Negative organisms: Escherichia coli (E coli), Gram-positive organisms: Staphylococcus aureus, Bacterial Culture: Bacterial cultures of Escherichia coli and Staphylococcus aureus were procured from Microbial Type Culture Collection (MTCC), Institute of Microbial Technology, Chandigarh.

Ans: appropriate corrections are made in the 3.8 and 2.5 sections.

6) Regarding biological part. Why were the MIC and MBC of the obtained silver nanoparticles not measured? What is the mechanism for improving the biological properties of AgNPs?

Ans: we carried out the MIC procedure:

Materials and methods

2.6. Minimum inhibitory concentration of silver nanoparticles

The determination of MIC of the AC- AgNPs against E.coli and S.auerus was performed as per the recommendations of the Clinical Laboratory Standard Institute (CLSI 2008b). The bacteria were grown under aerobic conditions at 37 °C for 24 h in LB broth. The cell count was made with a hemocytometer, and a standardized cell suspension (1×105 cells/mL) was prepared. One hundred microliters of the cell suspension were dispensed in triplicate microtiter wells to which 100 μL suspension containing different concentrations of nanoparticles in LB medium was added and mixed well. A conventionally used antibacterial agent such as triclosan [27] was used to compare the antibacterial potential of AgNPs. Triclosan was dissolved in dimethyl sulfoxide (DMSO) and diluted with sterile LB medium to obtain drug concentrations ranging from 0.0313 to 1,024 μg/mL. The cells with either nanoparticles or conventionally used antibacterial were incubated at room temperature for 24 h, and the growth inhibition was measured spectrophotometrically at 600 nm (Powerserve XS Biotech, USA). Wells with cells without AgNPs were used as the control. The MIC was calculated as the lowest amount of AgNPs that inhibited 50% growth of E.coli and S. aureus cells under the experimental conditions.

3.9. Minimum inhibitory concentration of silver nanoparticles

Minimum inhibitory concentration (MIC) test was performed to check the minimum concentration of biogenic AgNPs against bacterial strains. MIC test reveals the existence of antibacterial activity at different concentration levels. In the present investigation, the MIC test value of biogenic silver nanoparticles against both the strains of Escherichia coli MTCC40 pEx2717 and Staphylococcus aureus MTCC 87 Mkv2011 was calculated as 0.27, and 0.97 µg/mL, respectively and triclosan was calculated as 0.5 and 2µg/mL, respectively (Fig. 9). Interestingly the IC50 value was very much lower than the IC50 value of the commercial drug. AgNPs synthesized using different plant extracts were effective against various bacterias. For example, AgNPs biosynthesized by the Calotropis gigantea leaf extract against C. albicans with a MIC of 50 μg/mL [45]. In contrast, biosynthesized AgNPs using Artemisia annua showed MIC against C. albicans, C. tropicalis, and C. glabrata that ranged between 80 and 120 μg/mL. Thus, AgNPs produced by different approaches and species were reported to show antibacterial activity at different MIC levels depending on their size, shape, and surface modification [46].

The size of the nanoparticles plays an essential role in antimicrobial activity. It has been reported that the size and shape of metallic nanoparticles influence their chemical, optical, and thermal properties.

Figure 9. Minimum inhibitory concentration (MIC) of AC-AgNPs against Escherichia coli MTCC40 pEx2717 and Staphylococcus aureus MTCC 87 Mkv2011.

7) I believe that a comparison of the biological properties of silver nanoparticles without an organic component should be added (blank experiment).

Ans: Appropriate correction added as per reviewer 3's suggestions in biological properties of AgNPs

8) What about the toxicity of the obtained nanoparticles on normal cells?

Ans: Many researchers have shown that nanomaterials, including silver nanoparticles, can be lethal to cancer cells, unlike their macro counterparts, which remain neutral to cancerous changes. It has also been proven with in vitro studies that specific silver nanoparticles, in this case, silver nanoparticles  (AgNps), can act selectively on cancer cells, with significantly lower toxicity to normal cells –

Appropriate corrections are made in specific parts.

9) Minor comments.

- line 166. "Fig. 4 (a and b) “should be changed by "Fig. 3 (a and b) ".

Ans: The reference to the correct figure was corrected in the revised manuscript.

Round 2

Reviewer 1 Report

Accept as it

Author Response

The authors thank the reviewer very much for the time to review our manuscript and accept it as is.

Reviewer 2 Report

It wouls be good to show the effect of AgNPs on normal cells.

Why did you chose oflosacine and triclosan as a control for estimation of MIC50?

Author Response

Reviewer 2:

Q: It would be good to show the effect of AgNPs on normal cells.

Ans: Previous research on biosynthesized silver nanoparticles reported biocompatibility with normal cells. Researchers synthesized an AgNPs-loaded chitosan-alginate construct that interestingly showed excellent biocompatibility with normal cell line (L929) and cytotoxicity to cancer cells (HeLa cells) (Bilal et al (2017).

Inserted Ref at Page No:15 Section 3.10 : [53]. Bilal M, Rasheed T, Iqbal HMN, Li C, Hu H, Zhang X, Development of silver nanoparticles loaded chitosan-alginate constructs with biomedical potentialities. Int. J. Biol. Macromol, 105,393–400(2017).

Q: Why did you chose Ciprofloxacin and triclosan as a control for estimation of MIC50?

Ans: Separately answer for each compound are provided to reviewer 2, and the correspondent references are added to the manuscript to ease the potential readers' comprehension.

Ans: Ciprofloxacin, a second-generation quinolone, has a pH-dependent solubility (Jaman et al., 2015).

Inserted Ref at Page No:6 Section 2.5: [28] Jaman M, Chowdhury AA, Rana A A, Masum SM, Ferdous T, Rashid MA, Karim  MM,In vitro evaluation of ciprofloxacin hydrochloride. Bangladesh j. sci. ind. res. 50,251–256(2015). 

 Ans: Triclosan possesses broad-spectrum antimicrobial action and has been classified as a Class III drug (compounds with high solubility and low permeability) by FDA.

Inserted Ref at Page No: 6 Section 2.6 : [30]. Courtney KD, Moore JA, Teratology studies with 2,4,5-trichlorophenoxyacetic acid and2,3,7,8-tetrachlorodibenzo-P-dioxin. Toxicol. Appl. Pharmacol.  20, 396–403(1971).

Reviewer 3 Report

I thank the authors for answering my questions and improving the manuscript.

Author Response

Reviewer 3:

Comment: I thank the authors for answering my questions and improving the manuscript.

Ans: The authors thank the reviewer very much for the time to review our manuscript and accept it as is.

Round 3

Reviewer 2 Report

The introduction was revised. Highlights are meaningful. Other raised queries were also addressed.